# Effect of Ballroom Dancing on the Physical, Psychological, and Mental Well-Being of Oncological Patients: A Pilot Study

**DOI:** 10.3390/ijerph22040470

**Published:** 2025-03-21

**Authors:** Rebecca Schild, Martin Scharpenberg, Ivonne Rudolph, Jens Büntzel, Jutta Huebner

**Affiliations:** 1Klinik für Innere Medizin II, Universitätsklinikum Jena, Am Klinikum 1, 07747 Jena, Germany; jutta.huebner@med.uni-jena.de; 2Kompetenzzentrum für Klinische Studien Bremen, Fachbereich 3, Mathematik und Informatik, Hochschule Bremen, 28358 Bremen, Germany; mscharpenberg@uni-bremen.de; 3Waldburg-Zeil Kliniken, Rehabilitationsklinik Bad Salzelmen, Therapie/Leiterin Ambulante TherapieBadepark 5, 39218 Schönebeck, Germany; ivonne.rudolph@wz-kliniken.de; 4Klinik für HNO-Erkrankungen, Kopf-Hals-Chirurgie, Interdisziplinäre Palliativstation, Südharzklinikum, Robert-Koch-Straße 39, 99734 Nordhausen, Germany; jens.buentzel@shk-ndh.de

**Keywords:** neoplasm, ballroom dancing, dance sports, well-being

## Abstract

Purpose: Previous studies have shown that dancing can improve well-being, but few studies have looked at ballroom dancing. The present pilot study focuses on ballroom dancing and aims to investigate its effect on the physical, psychological, and mental well-being of cancer patients. Owing to COVID-19, face-to-face courses had to be replaced by online courses to continue the intervention and maintain learning progress; after the end of the pandemic, the courses could occur on site again when possible. Methods: For this cohort study, a total of 51 participants (38 patients and 12 healthy partners, no data for 1; 34 women and 9 men, no data for 8) participated. There were no limitations regarding the type of cancer, treatment, or comorbidities. It was an open access offering; participants were recruited through the newspaper and support groups. Using an anonymous standardized questionnaire and a numeric rating scale (NRS) ranging from 1 to 10, the participants were asked to rate their mental, physical, and psychological well-being at defined time points over one week. No side effects were registered and the teaching methods appeared to be practicable for the patients. Results: The pilot study showed an improvement in well-being after ballroom dancing. The results for physical, psychological, and mental well-being were significant (*p* < 0.0001). On average, men reported better scores, and all improvements were significant for both men and women. Overall, the healthy partners rated their well-being slightly better on average than the patients. A mixed-model analysis with repeated measurements in SAS was used to evaluate significant results (*p* < 0.05). Conclusion: Our data show that ballroom dancing for cancer patients has a positive effect on their well-being, even though their well-being returns to baseline levels within the following days. Positive effects were also demonstrated for participating healthy partners.

## 1. Introduction

Cancer treatment is often accompanied by sleep disturbances, fatigue, and physical and psychological stress, resulting in a decreased quality of life in concerned patients [1]. The positive effects of physical activity on these very symptoms have been demonstrated in several studies [2,3,4,5,6]. For example, a meta-analysis demonstrated that in 49,095 cancer patients, the disease recurrence rate was lower if they had a higher activity level [7]. In addition, a study conducted in Hungary with 55 cancer patients revealed that belly dancing as a complementary rehabilitation method led to an improvement in health-related quality of life [8]. In addition to physical benefits, dance therapy has also been shown to significantly improve psychological quality of life [9]. However, even independent of cancer treatment, physical inactivity can be causative for osteopenia, muscle loss, or obesity [10,11].

Dancing has only been studied in a small number of randomized controlled trials, where the types of dance, duration, and schedule varied [8,9,12,13,14]. The study by Szalai et al. investigated the quality of life and overall life satisfaction of patients with malignant diseases who attended belly dancing classes as an additional rehabilitation measure. It was found that quality of life increased as a result of the additional physical activity [8]. The study by Mannheim et al. also found a significant improvement in the quality of life of oncological patients who took dance classes during inpatient rehabilitation [9]. In 139 breast cancer patients, dance therapy reduced stress and pain during radiotherapy [12]. Overall, the literature shows that increased exercise leads to an improvement in the quality of life of cancer patients. Dancing appears to be a suitable form of physical activity for patients. In these studies, patients either participated as individuals or in a group. There are only few studies dealing with ballroom dancing. Owing to this lack of research, the present study aims to further investigate ballroom dances in cancer patients. Dancing may be associated with physical, mental, and social benefits, yet the data collected regarding these endpoints are heterogeneous. A simple approach to assessment had been developed due to the participants of the present study being reluctant to undergo a more extensive assessment and to maximize data collection.

In 2016, a private initiative by two of the authors (IR and JH) supported by a foundation started to offer ballroom dance classes for patients alone or with a partner. This initiative was one of the few to include partners of patients with cancer. Inviting partners together with patients may offer additional benefits for patients with respect to partnership, body image, and even sexuality. Owing to the wide range of dance styles, as well as the various music styles served by ballroom dancing, it is an activity that appeals to patients of all ages and activity levels. The different dance styles were also taught to prepare the participants to take part in other events, if desired, and to give them the opportunity to take further dance lessons after the course finished. The in-person dance program was interrupted by the COVID-19 pandemic. An online program was then offered for two years. The online course consisted of weekly lessons. The moves were taken from Standard and Latin American dances and adapted to limited space and dancing alone due to the pandemic situation. Most of the moves could be transferred to a couple of steps depending on the ability of the participants and the space available. The trainers were experienced ballroom dance instructors. All had previously worked with cancer patients in real-life classes. The online classes took place on a secure browser-based system that did not require a login password, just the link, which was the same for all of the classes in the course [15,16]. With the end of the pandemic, the face-to-face classes could be restarted again, and we planned to evaluate the effects on patients as well as their partners.

A previous pilot study confirmed the effectiveness of ballroom dancing in patients with different types of cancer receiving different treatments concerning their well-being [17]. This study investigated the influence of ballroom dancing on the well-being of cancer patients. A distinction is also made between physical, psychological, and mental well-being.

## 2. Materials and Methods

*Patient recruitment and lesson format.* For this pilot study, the results of a total of 51 participants were analyzed (Table 1): 38 were patients, 12 were healthy partners, while no data were given for 1 participant. In total, 34 women and 9 men were included, with the sex of 8 respondents was unspecified. With a total of 21 patients, those with breast cancer constituted the largest group. The sample size was not calculated in advance, participation was voluntary, and anyone who was interested could take part and was included in the study anonymously by returning their questionnaire. Participation was free of charge and participants learned about the offer through support groups and an advertisement in the local newspaper. Adults of any age and sex were welcome in the project. Patients with any type of cancer during or after therapy were allowed to participate. It was possible to participate with a partner who also had cancer, as well as with a healthy partner. Every week, a dance lesson of 90 min took place. One 10 min break was scheduled after 40 min, and individual pauses were possible at any time. The lessons were delivered by professional dance instructors who had been trained to understand the basic concepts of cancer diagnosis and treatment. They were trained to address the psychological and physical reactions of the patients to the diagnosis (e.g., fatigue and need for additional pauses, polyneuropathy and difficulty with steps, cognitive dysfunction, and difficulties with remembering the last lesson).

*Questionnaire.* The questionnaire used a numeric rating scale (NRS) ranging from 1 to 10 for well-being (1 is the highest level of well-being, and 10 is the lowest), which was divided into physical, psychological, and mental well-being, as previously described, and published by our group [17] (see Appendix A). The study by Schmidt et al. confirmed the effectiveness of an NRS-based method for assessing ballroom dancing in oncological patients. These patients had different types of cancer and were also undergoing different treatments. The simple approach of the NRS rating scale was easily accepted by the participants. For example, there is the ’Lebed method’, which can be used specifically for breast cancer patients to assess both their well-being and the range of motion of their arms. With the simple NRS approach, all participants who wish to take part can be approached and involved. This is an important aspect for a charity project, as in Germany there are many opportunities for dance events for breast cancer patients but only a few for prostate or bowel cancer patients, for example. As the questionnaire presented all scales for the different time points one below the other, the participants were provided with an overview of the evolution of their well-being. This direct feedback on their own condition may increase the patient’s motivation for the classes and documentation. Psychological well-being describes how participants feel, e.g., emotionally and socially, while mental well-being is be about the mind and thoughts but less about spirituality.

The participants were asked to document their well-being 3 days before and immediately before training, as well as immediately after, the evening after, and the following 6 days after training. Each participant was asked to return the questionnaire at the next session. The procedure was also repeated after 12, 24, and 48 weeks of follow-up courses. As a first analysis, several days after the training, quality of life returned to the preintervention level, and this was repeated at all four points in time. We decided to consider the measurements for every period as new measurements. Therefore, long-term effects were not part of the study

*Statistics.* The results of the questionnaires were first visualized with Excel and then analyzed via SAS Version 9.4. We fitted a linear mixed model with physical, psychological, and mental well-being as the outcome variables. Fixed effects were used for time and subgroup analysis for sex and patients or healthy partners, and a random intercept for follow-up courses was nested in individuals. The covariance structure of the random effects was compound symmetry. Well-being was compared between time points and subgroups via contrast tests and t-type confidence limits. *p* < 0.05 was considered significant.

*Ethical Approval.* The Ethical Committee of the University Hospital of Jena approved the study.

## 3. Results

*Demographic Data.* Among the 51 questionnaires distributed, 38 (74.5%) were completed by patients, and 12 (23.5%) were completed by their healthy partners. No data were given for 1 participant (2%). In total, 34 women and 9 men were included, with the sex of 8 respondents unspecified. A total of 13 participants (25.5%) had never attended a dancing lesson before, whereas 38 (74.5%) had some experience. Among the patients, those with breast cancer constituted the largest group (*N* = 21, 55.3%). Of the 51 participants, 23 danced with a partner (45.1%), 20 attended classes alone (39.2%), and 6 danced with family or friends (11.8%). (Table 1)

*Well-Being Before and After the Training*. As shown in Figure 1a–c, the mean values measured for mental, physical, and psychological well-being improved after the training. The mean value measured for physical well-being before training, at 4.85, was the worst of the three mentioned values. After the training, the value improved to 3.65 and stabilized at 4.30 after 4 days (*p* < 0.0001) (Table 2 and Table 3). Before the training, psychological well-being was the best at 4.45; in the evening after the training, the documented condition improved to 3.37; and even after 4 days, at 4.13, it was still slightly better than that before the training (*p* < 0.0001) (Appendix A). The value for mental well-being was in the middle of the range: it was 4.69 before training, improved to 3.73 during training, and then remained at an average value of 4.27 (*p* < 0.0001) (Appendix A).

*Women Compared to Men.* On average, men reported better scores in all three areas of well-being, and all improvements were significant for both men and women.

For men, physical well-being improved significantly from 4.36 to 3.14 (*p* < 0.0001) and remained slightly better than before training, at 3.94, after four days (*p* = 0.1119). The situation for women was similar, although less marked, with an improvement from 5.04 to 3.77 (*p* < 0.0001) and a score of 4.42 on day four (*p* < 0.0001) (Appendix A and Figure 2a). Psychological well-being was best, with men reporting an average score of 3.63 before training and women starting at 4.72. While the men improved to 2.79 (*p* = 0.0006), the women improved to 3.56 (*p* < 0.0001). Both groups had slightly improved scores at the end (men 3.68 (*p* = 0.85), women 4.43 (*p* = 0.0008)) (Appendix A and Figure 2b). Mental well-being followed a similar trajectory, with men improving from 4.21 to 3.26 (*p* = 0.0071) and then remaining at 4.05 (*p* = 0.5333), and women improving from 4.90 to 3.91 (*p* < 0.0001) and finally settling at 4.37 (*p* < 0.0001) (Appendix A and Figure 2c).

At first glance, the men gave better results than the women did, but the women’s sample was three times larger than the men’s sample, and there were more female patients in the women’s sample, while the men tended to participate as healthy partners (Appendix A). The differences between men and women were not significant.

*Patients Compared to Healthy Partners.* A comparison between healthy partners and patients also revealed that ballroom dancing improved physical, psychological, and mental well-being. For all three different types of well-being studied, these effects were significant for both partners and patients. The dynamics again showed that the improvement was greatest after training but then returned to the baseline in the following days. Some participants took part alone, others with a partner who was also suffering from cancer. Despite the limited number of healthy partners (*N* = 12), it was interesting to evaluate their scores, as they may also be heavily burdened, especially in terms of psychological and mental well-being.

Overall, on average, the healthy partners rated their physical, psychological, and mental well-being slightly better than the patients themselves. The difference in scores between patients and healthy partners was significant at baseline, but this effect was not maintained over time.

For physical well-being, the average score for the healthy participants was 4.25 at the start of the study, which improved to 3.45 after the training and decreased to 4.00 throughout the study (*p* = 0.28855). Patients, on the other hand, had worse average scores, starting at 5.01, rising to 3.70, and then to 4.38 (*p* < 0.0001) (Appendix A and Figure 3a).

The psychological well-being scores were best for the healthy partners. Well-being changed from 3.33 to the best-measured value of 2.92 (*p* = 0.0016), but a few days after the training, with a value of 3.63, it was even slightly worse than before the training—but this change could not be considered significant (*p* = 0.2855). Overall, the patients rated their mental well-being worse than the healthy partners. The initial score of 4.75 decreased to 3.49 (*p* < 0.0001) but remained at an average of 4.26 after four days (*p* < 0.0001) (Appendix A and Figure 3b).

The mental well-being scores were similar, with the healthy participants showing an average dynamic from 3.67 to 3.08 (*p* = 0.0086) and back to a score of 3.79 (*p* = 0.5725). The patients scored slightly lower on average, with an initial score of 4.96, which improved to 3.91 (*p* < 0.0001) after training and then returned to 4.39 (*p* < 0.0001) (Appendix A and Figure 3c).

*Adverse Events.* No adverse events occurred in any of the classes, and no cardiovascular problems, worsening of cancer treatment side effects (e.g., nausea), accidents, or musculoskeletal problems were observed.

## 4. Discussion

Our pilot study describes an improvement in well-being, mentally, physically, and psychologically, for both patients and healthy partners by participating in ballroom dancing classes. However, this improvement is temporary. Ballroom dancing was chosen for this study for several reasons. For example, ballroom dancing requires a partner: together, one is successful, and harmony between partners can enhance the effect, which may be an explanation for improvements in mental and psychological well-being. For example, in a study by Kunkel et al., 14 couples danced together, and those who had a good relationship with their partners reported more successful experiences and greater enjoyment when dancing [18]. In addition, ballroom dancing offers a wide variety of musical genres, movements, and dance styles, so it is more likely that the majority of participants in a class will find dances they like. Additionally, the intensity can be varied and adapted so that even weaker patients can participate. This is a special feature of classes for cancer patients because if individuals are exhausted, everyone understands because they can relate to the situation, either because of their own illness or their partner’s illness. These opportunities for reflection and adaptation may also have contributed to improved physical well-being. Additionally, many participants have taken dance classes in their youth, so one is often building on a foundation that has already been laid. In many cities, there are still dance schools, so there is a possibility to continue after the course.

An online investigation by Marschin et al. involved 365 healthy participants. They were asked about their exercise habits, which included statements about well-being and body image. Individuals who regularly participated in ballroom dancing stood out in the study: in terms of well-being, they gave the lowest scores on a psycho-diagnostic test (PHQ-2) and reported the most positive effects [19]. Several studies have focused on healthy subjects and dances, reporting on physical fitness and activity and their improvement. A study by Wang et al. revealed that 90 healthy Chinese students who participated in an eight-week jazz dance course were able to increase their fitness levels [20]. Another study in which middle-aged adults participated in a six-month dance course reported no significant effect on physical fitness [21]. The results concerning mental health and dance are also heterogeneous. A subjective improvement in participants’ self-esteem and cognitive function has been described, and Dwarika et al. reported that mental health in dance should be conceptualized as a comprehensive and dynamic state [22,23].

Overall, our pilot study revealed a significant positive effect (*p* < 0.001) of ballroom dancing on both patients and their healthy partners. The healthy participants scored higher on average in terms of well-being, but this may be due to the patients’ cancer, which can affect all three measurements. Patients are, of course, more likely to be physically impaired, to experience fatigue more quickly [24], or to feel more exhausted after exercise. Psychological stress often accompanies serious illness or affects patients not only during sports but also in everyday life. A study by Gregurek et al. described how cancer arouses more intense and emotional reactions than any other illness. Common accompanying psychological symptoms are fear of changes in body image, disability, and death, which are associated with psychological distress [25]. An important and newly found significant effect in our study is that partners also reported improvements in their physical, psychological, and mental well-being, which may contribute to healthy partners continuing to attend dance classes and maintaining motivation, which also contributes to an overall improvement in at least the psychological and mental well-being of patients. Overall, there were no major differences between the three levels of well-being for either the patients or the healthy partners, but in fact, psychological well-being scored slightly better.

Differentiating between women and men also shows that both groups studied documented a significant (*p* < 0.001) effect of ballroom dancing in terms of improving physical, psychological, and mental well-being. Over the centuries, it has been established that there are sports in the social imagination that have a more feminine connotation. Dance sports, in particular, tend to be associated with women, which has shaped the social image of women dancing rather than men [26]. In a study conducted in Germany in the early 2000s, dancing was ranked as the second most popular sport among adolescent girls, whereas dancing was of little or no interest to boys [27]. Contrary to the assumption that women might enjoy ballroom dancing more due to social circumstances, men reported better scores on average. However, this could also be because there were three times as many women as men in our sample, which may have confounded the comparison (Figure 2). On the other hand, the tasks of men and women in traditional ballroom dancing differ, with men (leaders) performing more decision-making tasks and leading, which might have additional effects. In a study by Vaczi et al., ten healthy amateur couples were studied during ballroom dancing. Physical parameters, such as peak heart rate and oxygen consumption, were measured, which revealed that women have a greater cardiovascular load because of their specific holding technique. This could also explain why men reported better overall well-being than women did [28].

To date, only a few studies on dancing with cancer patients have been published. In general, most projects in this field are open to female breast cancer patients. Owing to social gender roles and socialization, it is still assumed that women may be more likely to dance, which makes it appropriate to study breast cancer patients in relation to this form of therapy, but we wanted to address and include all types of cancer, as well as their partners.

In 2017, a randomized trial of ballroom dancing published by Pisu et al. reported a significant positive effect on patients’ quality of life (QOL) (physical activity, mental component of QOL, vitality) [29]. This study did not find a benefit for partners, but our study did, regarding improved well-being. One reason for this may be that healthy partners attend dance classes for their ill partner’s enjoyment and may be less interested in themselves. In the study by Pisu et al., participants were asked to train for the course several times a week, even in their free time, which may have reduced the willingness and enjoyment of the healthy participants.

What distinguishes our study from other published studies on dance and cancer patients is that we offered ballroom dance classes, not Dance Movement Therapy (DMT). DMT is generally offered more often. Various concepts and methods of depth psychology and behavioral therapy are used to treat patients psychologically and physically. The American Dance Therapy Association (ADTA) defines DMT as the psychotherapeutic use of movement to promote the emotional, social, cognitive, and physical integration of individuals with the goal of improving health and well-being. Dance is recognized as a form of psychotherapy because the body and mind are seen as inextricably linked. Movement is used not only as an intervention but also as an assessment tool. Dance/movement therapists assess the client’s movements and observe and evaluate their own movements. They use verbal and nonverbal communication to develop and implement interventions that address the individual’s emotional, social, physical, and cognitive integration [30]. For example, Sandel et al. published a study about DMT in 2005 and reported that the Lebed Method improved the range of motion of the arms and patients’ QOL [31]. In our project, we did not intend to offer dance therapy in this sense, but rather ballroom dance classes in which common dance steps are demonstrated and practiced, as they may still be known from dance classes of youth. The dancing and movement of the participants were not assessed, and no conclusions were drawn from the execution of the dance steps. Only at defined points in time, which were not during the intervention, were participants asked to rate how their physical, psychological, and mental well-being had changed.

It would be interesting to learn about the long-term effects of ballroom dancing on both patients and healthy partners. Perhaps this can lead to long-term improvement in general well-being, as longer training may also lead to confidence and a certain routine of dance steps; thus, more difficult dance steps may be possible, which could certainly improve self-confidence and thus psychological and mental well-being. Regular training and movement of the whole body could also lead to a lasting improvement in physical well-being.

With respect to the reliability of the data, several limitations of the study should be noted. The sample size was small overall and also had high heterogeneity, e.g., different types of cancer and a wide age range (Table 1). Also, it was a convenience sample; the participants self-selected to join the intervention. A control or comparison group was missing, and it was not possible to blind the participants to the study. In addition, well-being was assessed via only one single item, the VAS, but its feasibility has been proven in a previous study. Direct feedback and an overview of one’s own development by completing questionnaires can increase the motivation of the participants and thus keep the dropout rate low. The questionnaire also did not cover other factors that may have an additional influence on well-being, such as diet, other physical activities and sports, or other social factors that may influence an individual’s daily well-being. Larger studies are needed to determine the acceptability of this therapy as part of the continuum of care for cancer patients and survivors.

## 5. Conclusions

In conclusion, the data of our pilot study show that even if well-being returns to baseline levels a few days after training, dance classes for cancer patients have a positive effect on their well-being. There is a need for a more controlled study that is able to control for potential confounders as well as any effect modifiers that influence the results of the study. In further studies, it may be useful to compare other physical activities with ballroom dancing to determine whether there are differences in feasibility and outcomes in terms of well-being. Nevertheless, dancing seems to be a valuable and feasible form of physical activity for cancer patients and survivors and has a positive effect on patients and even their healthy partners.

## Figures and Tables

**Figure 1 ijerph-22-00470-f001:**
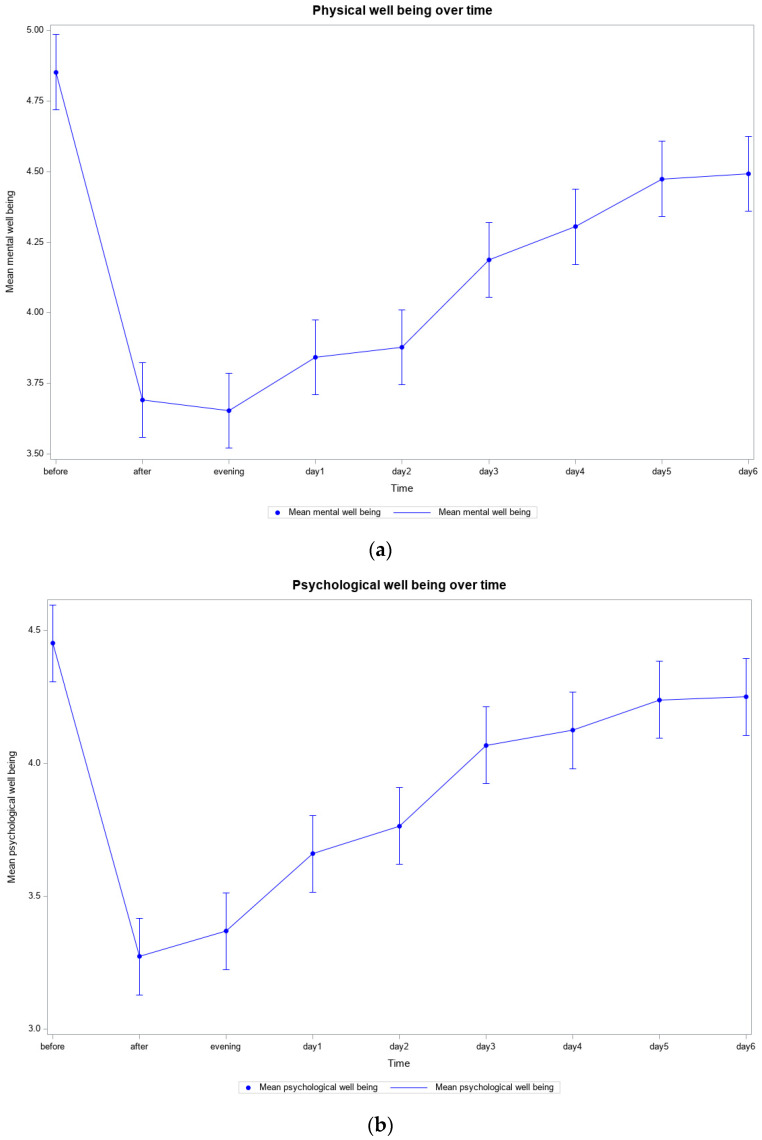
(**a**) Physical well-being of all participants over time (*N* = 51); (**b**) Psychological well-being of all participants over time (*N* = 51); (**c**) Mental well-being of all participants over time (*N* = 51).

**Figure 2 ijerph-22-00470-f002:**
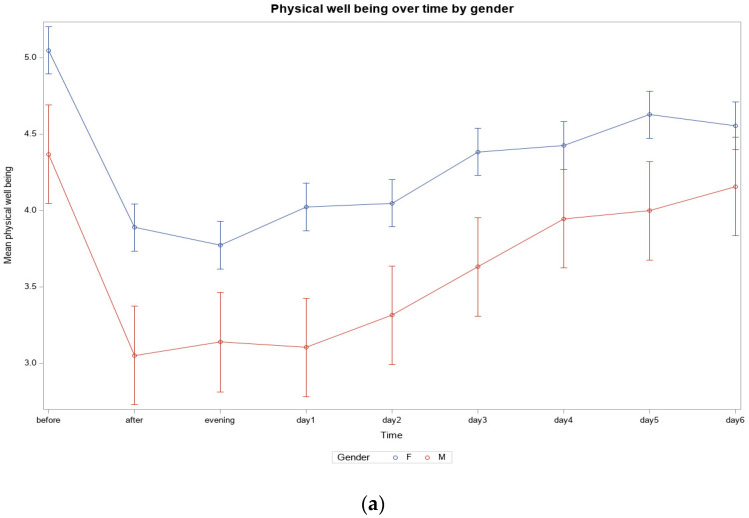
(**a**) Physical well-being over time by sex (*N* = 43, Women = 34, Men = 9; (**b**) Psychological well-being over time by sex (*N* = 43, Women = 34, Men = 9; (**c**) Mental well-being over time by sex (*N* = 43, Women = 34, Men = 9).

**Figure 3 ijerph-22-00470-f003:**
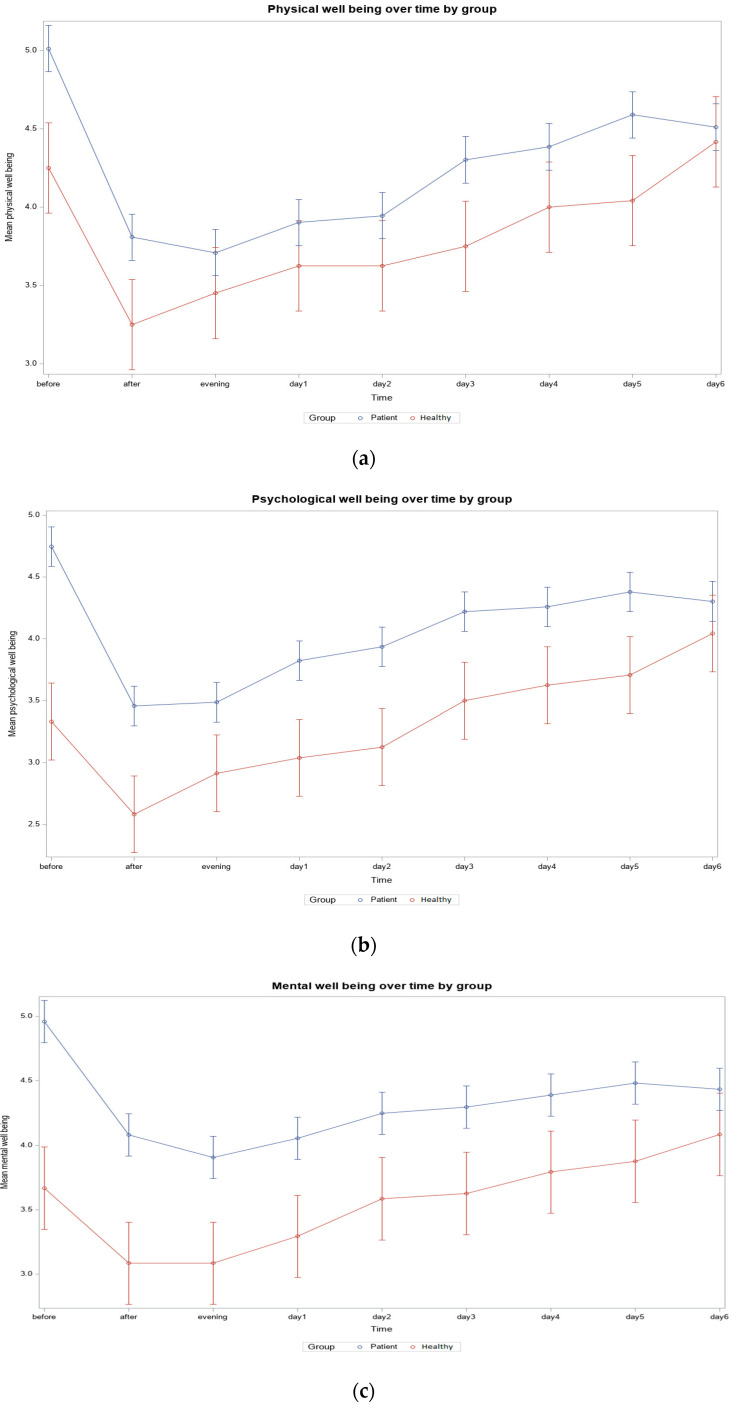
(**a**) Physical well-being over time: Patient vs. Healthy Partner (*N* = 50, patient = 38, healthy = 12); (**b**) Psychological well-being over time: Patient vs. Healthy Partner (*N* = 50, patient = 38, healthy = 12); (**c**) Mental well-being over time: Patient vs. Healthy Partner (*N* = 50, patient = 38, healthy = 12).

**Table 1 ijerph-22-00470-t001:** Demographic data, *N* =51.

Data	No.	%
*Age, years*		
<30	0	
31–40	3	5.9
41–55	11	21.6
55–65	22	43.1
66–75	12	23.5
>75	2	3.9
No Data	1	2
*Sex*		
Women	34	66.7
Men	9	17.6
No Data	8	15.7
*Status*		
Patient	38	74.5
Healthy Partner	12	23.5
No Data	1	2
*Type of Cancer*		
Breast	21	55.3
Skin (Melanoma)	4	10.5
Uterus	3	7.9
Ovarian	2	5.3
Stomach	1	2.6
Tubes	1	2.6
Chondrosarcoma	1	2.6
Colorectal	1	2.6
Pancreas	1	2.6
Ear-Nose-Throat	1	2.6
No Data	2	5.3
*Time since first diagnosis, years*		
1	1	
2	11	
3	13	
4	11	
No Data	2	
*Current Cancer Therapy*		
Yes *	18	47.4
No	18	47.4
No Data	2	5.3
*Previous dance Experience*		
Yes	38	74.5
No	13	25.5
No data	0	
*Participation together with*		
Partner	23	45.1
Friend	1	2
Family	5	9.8
Alone	20	39.2
No Data	2	3.9

* radiotherapy *N* = 4, other cancer drugs *N* = 11, chemotherapy *N* = 5, endocrine therapy *N* = 9, other *N* = 4.

**Table 2 ijerph-22-00470-t002:** Comparison of mean physical well-being of all participants (*N* = 51).

Comparison	Mean Difference	Standard Error	*p*-Value	Lower 95% Confidence Bound	Upper 95% Confidence Bound
Before—after	1.1609	0.1070	<0.0001	0.9510	1.3708
Before—evening	1.1984	0.1072	<0.0001	0.9879	1.4088
Before—day1	1.0087	0.1070	<0.0001	0.7988	1.2186
Before—day2	0.9739	0.1070	<0.0001	0.7640	1.1838
Before—day3	0.6652	0.1070	<0.0001	0.4553	0.8751
Before—day4	0.5476	0.1075	<0.0001	0.3367	0.7586
Before—day5	0.3779	0.1072	0.0004	0.1675	0.5884
Before—day6	0.3604	0.1072	0.0008	0.1499	0.5708

**Table 3 ijerph-22-00470-t003:** Means of physical well-being of all participants (*N* = 51).

Time	Mean	Standard Error	*p*-Value	Lower 95% Confidence Bound	Upper 95% Confidence Bound
before	4.8522	0.1326	<0.0001	4.5920	5.1123
after	3.6913	0.1326	<0.0001	3.4312	3.9515
evening	3.6538	0.1328	<0.0001	3.3932	3.9144
day1	3.8435	0.1326	<0.0001	3.5833	4.1036
day2	3.8783	0.1326	<0.0001	3.6181	4.1384
day3	4.1870	0.1326	<0.0001	3.9268	4.4471
day4	4.3045	0.1330	<0.0001	4.0435	4.5655
day5	4.4742	0.1328	<0.0001	4.2137	4.7348
day6	4.4918	0.1328	<0.0001	4.2312	4.7524

## Data Availability

The analyzed datasets generated during the study are available from the corresponding author upon reasonable request.

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
