# Peer review of "Effect of Ballroom Dancing on the Physical, Psychological, and Mental Well-Being of Oncological Patients: A Pilot Study"

_ijerph, 2025, doi:10.3390/ijerph22040470_

Round 1

Reviewer 1 Report (Previous Reviewer 1)

Comments and Suggestions for Authors

The content of the manuscript is acceptable, and the authors made the appropriate modifications from the previous comments.  However, I do believe it needs to be copy edited to modify any grammatical and formatting errors. 

Comments on the Quality of English Language

The manuscript needs to be copy edited by a professional. 

Author Response

We  thank the reviewer for this comment. A professional has revised the text and we have also run an AI over the text. Not all the changes were marked in the text, otherwise it would have been too confusing.

Reviewer 2 Report (Previous Reviewer 3)

Comments and Suggestions for Authors

·         The purpose in the abstract has to be rewritten for clarity

·         Line 16-17 – this sentence is irrelevant in the purpose section

·         Line 18- 19 – the description of the participants is not clear. Please clarify the number of participants and present the breakdown of the participants and gender more clearly

·         Line 21- what is daily paper – newspaper?

·         The result section in the abstract doesn’t have the results!

·         Line 50-51 – is it on cancer patients? what is the findings of these studies

·         Line 55- 57 – how this paragraph related to the study

·         The introduction is poorly written. There is no adequate background information to support the rationale of the study, and the rationale also not properly presented. What is ballroom dancing and how is it different from other forms of dancing? Are there any previous studies conducted on the effectiveness of this type of dance form?

·         Line 68- 76- this should not be in the introduction

·         Line 86- this should be in the results

·         Lack of sample size calculation is a major limitation of the study.

·         The methodology is poorly written and not improved in the updated version. The procedure is not clear. I have concerns regarding the questionnaire used in the study. What are the outcome measures used? What is the reliability and validity of these outcome measures?

·         There are repeated measurements of outcome measures at various intervals including 12,24 and 48 weeks. However, the results are not reported in the results section

·         What is the purpose of data collection three days before and immediately before the intervention

·         Why repeated measure Anova not used for statistical analysis

·         Only physical well-being data is reported in tables. What about other data. It is highly recommended to report in tables than graphs

·         There is no critical analysis of the results in the discussion section . Discussion looks like a literature review

·         There are number of confounding factors and there is no control group in the study 

Author Response

  1. The purpose in the abstract has to be rewritten for clarity

Answer:  We thank the reviewer for the comment. We have revised the abstract to make the aim of the study clearer.

  1. Line 16-17 – this sentence is irrelevant in the purpose section

Answer:  We thank the reviewer for the comment and have adjusted the purpose in the abstract to make it clearer. In a previous round of review, a reviewer asked that the information about the online courses during Covid 19 be included and briefly described in the abstract, so we would like to keep this information in the abstract.

  1. Line 18- 19 – the description of the participants is not clear. Please clarify the number of participants and present the breakdown of the participants and gender more clearly

Answer: We thank the reviewer for pointing this out. We have described the number of participants, as well as the breakdown into patients and healthy partners, and men and women. (a total of 51 participants (38 patients and 12 healthy partners, no data for 1; 34 women and 9 men, no data for 8) participated)

  1. Line 21- what is daily paper – newspaper?

Answer:  We thank the reviewer for pointing this out and we changed “daily paper” to “newspaper”.

  1. The result section in the abstract doesn’t have the results!

Answer:  We thank the reviewer for pointing this out. We have added the comparison between men and women and the comparison between healthy partners and patients to the results section of the abstract.

  1. Line 50-51 – is it on cancer patients? what is the findings of these studies
  2. The introduction is poorly written. There is no adequate background information to support the rationale of the study, and the rationale also not properly presented. What is ballroom dancing and how is it different from other forms of dancing? Are there any previous studies conducted on the effectiveness of this type of dance form?

Answer: We thank the reviewer for pointing this out; we have described the studies in the introduction in more detail and mentioned that increased physical activity through dance has already been shown to have a positive effect on the quality of life of cancer patients in other studies to create more background.

  1. Line 55- 57 – how this paragraph related to the study

Answer:  We would like to thank the reviewer for the comment. this sentence in the introduction is intended to make it clear that there are only a few studies on oncological patients who dance ballroom dancing. In order to generate a higher compliance, it was necessary to use a simple examination tool, for this purpose we designed an easy-to-understand questionnaire, for which another pilot study was carried out, which we also describe in the article.

  1. Lack of sample size calculation is a major limitation of the study.

Answer:  We thank the reviewer for pointing this out. We are aware that this is a major limitation of the study. We have included this in our limitations, in a further study the sample size needs to be calculated.

  1. The methodology is poorly written and not improved in the updated version. The procedure is not clear. I have concerns regarding the questionnaire used in the study. What are the outcome measures used? What is the reliability and validity of these outcome measures?

  1. Why repeated measure Anova not used for statistical analysis

  1. There are repeated measurements of outcome measures at various intervals including 12,24 and 48 weeks. However, the results are not reported in the results section

Answer:  We thank the reviewer for these comments. Linear mixed effect models were used instead of repeated measures ANOVA as they allow for missing data (which were given at some time points) and more flexible modeling. In order to avoid power loss and bias introduced by omiting data from individuals with partially missing data in a repeated measures ANOVA, and in order to answer the study's question, linear mixed models were chosen.

In line 127-130 we describe that long-term effects were not part of the study: “As a first analysis, several days after the training, quality of life returned to the preintervention level, and this was repeated at all four points in time. We decided to consider the measurements for every period as new measurement. Therefore, long-term effects were not part of the study“.

  1. What is the purpose of data collection three days before and immediately before the intervention

Answer: We thank the reviewer for pointing this out. We asked the participants to document their well-being 3 days before and immediately before the intervention in order to have a stable baseline value

  1. Only physical well-being data is reported in tables. What about other data. It is highly recommended to report in tables than graphs

Answer:  We thank the reviewer for the comment. We wanted to show all the graphs in the article because they are quick and easy to interpret. We have not included all the tables to avoid making the article too long, so we have described the results in the main text but included the tables in the appendix.  We focused on physical well-being as ballroom dancing is a sport/dance style and we thought this would be the most interesting component. 

  1. There is no critical analysis of the results in the discussion section. Discussion looks like a literature review

Answer:  We thank the reviewer for this comment. In the discussion we try to review our own results and discuss, for example, the comparison between participating men and women and between healthy partners and patients. We have tried to use recent data as references. We have also tried to define the limitations of the study in a structured way and to elaborate on them based on the editor's comments. Since the other reviewers only criticised the discussion a little, we would be grateful if the reviewer would be so kind to describe in more detail what we could improve in the discussion.  

  1. There are number of confounding factors and there is no control group in the study 

Answer:  We thank the reviewer for his comments. We have self-critically analysed the confounding factors and listed them in the limitations of our study.

Reviewer 3 Report (New Reviewer)

Comments and Suggestions for Authors

Authors' review report attached

Author Response

  1. The title, although somewhat long, is appropriate in relation to the content of the study. As a suggestion, I would change the phrase “patients with cancer” to “oncological” patients.

Answer:  We thank the reviewer for the comment and changed the title to “Effect of ballroom dancing on the physical, psychological and mental well‐being of oncological patients- a pilot study”.

  1. The abstract follows a suitable structure, but the objective is not clearly defined. The various corrections made in the text cause confusion when reading, making it unclear which part of the text is correct and which has been modified. Despite this, the objective of the study is still not clear. The abstract is slightly long, and I recommend omitting data such as: the characteristics of the participants, the statistical analysis data or the assessment scale. I Would recommend including in the abstract the name of the instrument used.

Answer:  We thank the reviewer for the comments. We have now incorporated the changes to make it clearer to read. We have reworded lines 14-15 to make the aim of the study clearer.

The length of the abstract increased after the first round of review. One reviewer asked for additional information about the participants (gender, number of healthy partners, patients), so we would like to leave this information in the text.

  1. The article presents a simple introduction, with slight scientific evidence about ballroom dancing as an adjuvant treatment to medical treatment processes against cancer. As background for this article, I recommend discussing the multiple pieces of publishing evidence concerning the impact of physical activity on the quality of life of patients undergoing oncological treatment. The introduction ends with the objective of the research, which is defined very briefly as “to investigate the influence of the ballroom dancing on the wellbeing of cancer patients”.

Answer:  We thank the reviewer for pointing this out; we have described the studies in the introduction in more detail and mentioned that increased physical activity through dance has already been shown to have a positive effect on the quality of life of cancer patients in other studies to create more background.

  1. With the regard to the information in Section 2, Material and Methods, specifically in the paragraph about patient recruitment, very little is explained about the characteristics of the participants, whereas in the results section this information is explained more thoroughly. I recommend doing it reverse. In this section, the objective is repeated once again.

Answer:  We thank the reviewer for pointing this out, and we have added more participant information to the Materials and Methods section.

  1. Regarding the research instruments, the instrument is not defined, and there is only a reference to a previous publication by the group. I recommend presenting and explaining the instrument used for data collection so that the reader can more easily understand and comprehend the tool. The statistical process used is well described.

Answer:  We thank the reviewer for this comment. We describe our questionnaire in lines 104 - 121. It was not a standardised questionnaire, but one developed by our group. We tested the feasibility of this questionnaire in a previous pilot study. We have included a version of the questionnaire as an esupplement in the appendices

  1. As for the results obtained, the data are presented clearly. The tables used for this (Table 2 and 3) are easy to interpret and understand, but they only show the physical wellbeing data, while the data on mental and psychological wellbeing are included in the appendices. Why have the physical data been given more value? I recommend combining all of them and presenting them appropriately.

Answer: We thank the reviewer for the comment. We wanted to show all the graphs in the article because they are quick and easy to interpret. We have not included all the tables to avoid making the article too long, so we have described the results in the main text but included the tables in the appendix.  We focused on physical well-being as ballroom dancing is a sport/dance style and we thought this would be the most interesting component.

  1. In terms of the discussion and conclusions, the article is adequate and features current works that reinforce the conclusions and the data obtained while also defining the study’s limitations.

  1. The references used throughout the article are current, especially in the discussion, with a large predominance of references from the last 7 years, which is appreciated and confirms the appropriateness of their use. The same applies to the number of self-citations by the authors, which is very low.

Answer:  We thank the reviewer for their valuable recommendations and the attached notes. We have revised the manuscript thoroughly and hope that it is now suitable for publication

Round 2

Reviewer 2 Report (Previous Reviewer 3)

Comments and Suggestions for Authors

  • The manuscript is improved substantially. I have  few more comments
  • The result section in the abstract is written too generally. Was there any statistically significant difference after the intervention?
  • The conclusion in the abstract is just a repetition of the results.
  • I have concerns regarding the questionnaire used in the study. What is the reliability and validity of the NRS well-being scale?

Author Response

We thank the reviewer for the valuable comments and would like to respond to the remaining questions.

  1.  The result section in the abstract is written too generally. Was there any statistically significant difference after the intervention?

  1. The conclusion in the abstract is just a repetition of the results.

Answer: We have revised both the results and the conclusion in the abstract. We have included the statistical values in the results and reworded them so that they are not directly repeated.

  1. I have concerns regarding the questionnaire used in the study. What is the reliability and validity of the NRS well-being scale?

Answer: We thank the reviewer for pointing this out. To confirm the effectiveness of an NRS-based assessment, our group conducted a pilot study in 2018. To remove any confusion, we have explained in more detail what this study was about under 'Questionnaire'.  (line 129-138)

This manuscript is a resubmission of an earlier submission. The following is a list of the peer review reports and author responses from that submission.

Round 1

Reviewer 1 Report

Comments and Suggestions for Authors

Overall, the paper has merit. With the Arts in Health movement, dance therapy is a growing area. Studies such as this can be helpful and should be published. Below are relatively minor issues I think should be addressed:

Line 15, can the authors define the participants by sex and race here? It would be helpful for the reader.

Line 19, a comma should separate registered and and.

Line 23, delete "In conclusion..."

Line 34, for reference number 7, was dance one of the activity levels measured? If so, that would be great to discuss or list here. If not, perhaps this sentence could be massaged a bit to include another study that measures dance in METs or something similar. I do not know this literature well, but I suspect the intensity of activity may be important, too. Perhaps the authors could address this in a sentence or two.

Line 44, the comma between developed and due should be a period. Or, the authors could add "and" after the comma to form a complex sentence.

Lines 58-61, could the authors expand these two, one-sentence paragraphs? The information is important, and I think expanded these would be helpful to the reader.

Line 64, I suspect the authors actually are referring to sex and not gender. Sex is biological; gender is sociological. I would recommend the authors replace gender with sex in their paper, as I do not believe they are studying sociological traits. This can be a tricky issue in today's world, so I think erring on the biological aspect is better.

Line 66, can the "pause" be better defined with respect to when it occurs, how long it lasts, etc. 

Line 85, a period instead of a comma would be better between time and we.

Line 99, replace the comma with a period between partners and no.

Line 100, I would recommend using men and women instead of male and female, as the authors are working with humans. It is fine to use male and female as adjectives, e.g., female cancer patient, but many view it as impersonal to describe a person as male or female instead of a man or woman. It is just a suggestion, but I think it would be better to do this.

Line 115, female vs. male is probably better women vs. men. I am not sure I would write vs. Perhaps rewording this as a comparison and not a competition or an adversarial situation. 

Line 116, add a comma between being and and.

Table 1, replace Gender with Sex, and replace Female and Male with Women and Men. Is the skin cancer melanoma? I suspect it is as the other forms of skin cancer generally are treated rather easily with cryotherapy or excision. If it is melanoma, I would recommend writing: Skin (Melanoma).

Line 166, did the participants just attend the courses? Would participating be better? It is the activity that is improving the health of the participants.

Lines 168 and 169, can the authors cite the sources for these statements? I do not doubt the truthfulness, but a source or two would be helpful. 

Lines 174 and 175, when using someone or everyone, it is better to avoid using a plural pronoun such as "they." I would suggest rewriting this to avoid any confusion. The easiest way is to replace someone with individuals and then change the verb from is to are. 

Line 178, using "you" is awkward and informal. Use one instead.

Line 183, persons or individuals would be better than people. 

Line 186, is its correct? The authors are referring to two things, i.e., fitness and activity, so their would be better.

Line 197, while this seems obvious, I believe a source should be cited.  Also, this might need to be expanded a bit to make the "why" more specific to the reader.

Lines 209 and 210, numerous reasons exist why women might dance more than men. I would recommend this section be expanded and cited. The assumption about women and then the scores about men need more explanation.

Line 242, the authors need to explain why "women may be more likely to dance" for social reasons. I am not disagreeing with this, but more explanation and a few references need to be cited. It is well documented in the literature that collegiate physical education programs, especially for women, involved dance, especially in the early part of the 20th century (in America, for sure). I am not sure about Europe, but this should be discussed and expanded. 

Line 259, the issue of dance therapy and ballroom dance needs to be explained a bit more. 

The article needs a good copy editor for English, but only for "polishing" the English rather than rewriting it. 

Comments on the Quality of English Language

See above.

Author Response

We thank the reviewers for their valuable recommendations and the attached notes. We have revised the manuscript thoroughly and hope that it is now suitable for publication

Reviewer 1 Report:

Overall, the paper has merit. With the Arts in Health movement, dance therapy is a growing area. Studies such as this can be helpful and should be published. Below are relatively minor issues I think should be addressed:

  1. Line 15, can the authors define the participants by sex and race here? It would be helpful for the reader.

Answer:

We thank the reviewer for the comment and positive assessment of our study. Furthermore, we agree with the reviewer, so we have added a subdivision into sex, a subdivision into race was not included in our study

  1. Line 19, a comma should separate registered and and.

Line 23, delete "In conclusion..."

Answer:

We thank the reviewer for the comment and have taken up the corrections

  1. Line 34, for reference number 7, was dance one of the activity levels measured? If so, that would be great to discuss or list here. If not, perhaps this sentence could be massaged a bit to include another study that measures dance in METs or something similar. I do not know this literature well, but I suspect the intensity of activity may be important, too. Perhaps the authors could address this in a sentence or two.

Answer:

We thank the reviewer for the comment and have included another study and described it in more detail.

  1. Line 44, the comma between developed and due should be a period. Or, the authors could add "and" after the comma to form a complex sentence.

Answer:

We thank the reviewer for the comment and have taken up the corrections

  1. Lines 58-61, could the authors expand these two, one-sentence paragraphs? The information is important, and I think expanded these would be helpful to the reader.

Answer:

We thank the reviewer and have described in more detail the interruption at COVID times and the online course offering that followed. We also added another table in the supplements that shows the concept of the online lessons, that our trainers were working with.

  1. Line 64, I suspect the authors actually are referring to sex and not gender. Sex is biological; gender is sociological. I would recommend the authors replace gender with sex in their paper, as I do not believe they are studying sociological traits. This can be a tricky issue in today's world, so I think erring on the biological aspect is better.

Answer:

We absolutely share the opinion of the reviewer and have revised the entire paper and replaced ‘gender’ with ‘sex’. It is absolutely correct that there is a sociological and biological difference, which is important to note. In our text we refer to the biological context

  1. Line 66, can the "pause" be better defined with respect to when it occurs, how long it lasts, etc. 

Answer:

We thank the reviewer for the comment and have added more detail. One pause of 10 min was scheduled for the participants at half time. If they needed more time to relax, or needed individual pauses, they were free to take breaks at any time.

  1. Line 85, a period instead of a comma would be better between time and we.

Line 99, replace the comma with a period between partners and no.

Answer:

We thank the reviewer for the comment and have taken up the corrections

  1. Line 100, I would recommend using men and women instead of male and female, as the authors are working with humans. It is fine to use male and female as adjectives, e.g., female cancer patient, but many view it as impersonal to describe a person as male or female instead of a man or woman. It is just a suggestion, but I think it would be better to do this.

Answer:

We thank the reviewer for the comment and have also revised the paper here, female has been replaced by women, male by men

  1. Line 115, female vs. male is probably better women vs. men. I am not sure I would write vs. Perhaps rewording this as a comparison and not a competition or an adversarial situation. 

Answer:

We share the opinion of the reviewer, this is not a competition but merely a comparison, so the wording has been adapted

  1. Line 116, add a comma between being and and.

Answer:

We thank the reviewer for the comment and have taken up the corrections

  1. Table 1, replace Gender with Sex, and replace Female and Male with Women and Men. Is the skin cancer melanoma? I suspect it is as the other forms of skin cancer generally are treated rather easily with cryotherapy or excision. If it is melanoma, I would recommend writing: Skin (Melanoma).

Answer:

We thank the reviewer for the comment and have taken up the corrections

  1. Line 166, did the participants just attend the courses? Would participating be better? It is the activity that is improving the health of the participants.

Answer:

We thank the reviewer for the comment and have taken up the corrections

  1. Lines 168 and 169, can the authors cite the sources for these statements? I do not doubt the truthfulness, but a source or two would be helpful. 

Answer:

We thank the reviewer for the comment and have added a source that confirms the statement

  1. Lines 174 and 175, when using someone or everyone, it is better to avoid using a plural pronoun such as "they." I would suggest rewriting this to avoid any confusion. The easiest way is to replace someone with individuals and then change the verb from is to are. 

Answer:

We thank the reviewer for the comment and have taken up the corrections

  1. Line 178, using "you" is awkward and informal. Use one instead.

Line 183, persons or individuals would be better than people. 

Line 186, is its correct? The authors are referring to two things, i.e., fitness and activity, so their would be better.

Answer:

We thank the reviewer for the comment and have taken up the corrections

  1. Line 197, while this seems obvious, I believe a source should be cited.  Also, this might need to be expanded a bit to make the "why" more specific to the reader.

Answer:

We thank the reviewer for his comment and have expanded the ‘why’ and added a source. We hope it is now more specific and easier to understand.

  1. Lines 209 and 210, numerous reasons exist why women might dance more than men. I would recommend this section be expanded and cited. The assumption about women and then the scores about men need more explanation.

Line 242, the authors need to explain why "women may be more likely to dance" for social reasons. I am not disagreeing with this, but more explanation and a few references need to be cited. It is well documented in the literature that collegiate physical education programs, especially for women, involved dance, especially in the early part of the 20th century (in America, for sure). I am not sure about Europe, but this should be discussed and expanded.

Answer:

We thank the reviewer for the comment and share this opinion. We tried to address the social aspects of why women might like dancing more than men and included relevant sources in the paper.

  1. Line 259, the issue of dance therapy and ballroom dance needs to be explained a bit more. 

Answer:

We thank the reviewer for pointing this out, we described dance movement therapy in more detail and what makes it different from our intervention. 

  1. The article needs a good copy editor for English, but only for "polishing" the English rather than rewriting it. 

Answer:

We thank the reviewer for this comment. To correct the English a little, we had the text edited dy Springer’s AI ‘Curie’. 

Reviewer 2 Report

Comments and Suggestions for Authors

It's a well-written article and such an interesting research.

The figures and tables since they are pretty clear. On the other hand, I had really serious concerns about the actual impact of this paper on society. The protocol applied by the authors encompasses 1 week follow-up with a 90 min dance training delivered online. Given that to stabilise a certain skill or develop another physical or psychological characteristic it might take months or even years, the authors say that they are not investigating the long-term effects. Still, they highlight that the patient and their partner had psychological and physical benefits after 1 training of 90 minutes, including a break, delivered online. Maybe what could help their work to make an impact on the real – world would be structuring a practical guide explaining in detail the dance training they taught, so that other researchers could develop this protocol. I recommend to the authors to give specific guidelines on how to implement an online dance training to this population. In particular, they could make a table, writing the specific exercise they taught, how they managed the break (after a certain exercise, before teaching new techniques, etc…), how they gave feedback online and how to implement such a lesson with beginners.

Author Response

It's a well-written article and such an interesting research.

The figures and tables since they are pretty clear. On the other hand, I had really serious concerns about the actual impact of this paper on society. The protocol applied by the authors encompasses 1 week follow-up with a 90 min dance training delivered online. Given that to stabilise a certain skill or develop another physical or psychological characteristic it might take months or even years, the authors say that they are not investigating the long-term effects. Still, they highlight that the patient and their partner had psychological and physical benefits after 1 training of 90 minutes, including a break, delivered online. Maybe what could help their work to make an impact on the real – world would be structuring a practical guide explaining in detail the dance training they taught, so that other researchers could develop this protocol. I recommend to the authors to give specific guidelines on how to implement an online dance training to this population. In particular, they could make a table, writing the specific exercise they taught, how they managed the break (after a certain exercise, before teaching new techniques, etc…), how they gave feedback online and how to implement such a lesson with beginners.

Answer:

We thank the reviewers for their valuable recommendations and the attached notes. We have revised the manuscript thoroughly and hope that it is now suitable for publication.

We thank the reviewer for pointing this out and agree that further research is needed in the future to find out about the long-term effects. We would like to thank you for the suggestion to create a table with specific guidelines so that courses can be continued, especially online. We have summarized the concept of our online course in a table, which we would like to add to the article.

Here, together with our trainer, we worked out how certain learning content could also be conveyed online.

Reviewer 3 Report

Comments and Suggestions for Authors

The title is too general. What is meant by well-being?

The conclusion is written as background in the abstract

Line 14 – How is this background related to the study?

The methodology is not clear in the abstract.

Line 33- 34- this statement is about cancer survivors. How is it relevant here?

Line 36-38 -meaning not clear

There is no supporting literature in the introduction. Most of the sentences written in the introduction is not relevant to the study. In addition, the rationale, the research gap and the aim of the study is not stated.

The methodology is poorly written in is too brief. What is the design of the study? How were the participants recruited? How is the sample size calculated? How many participants were recruited? How the data collection was performed. What is the reason for recruiting healthy participants? How were they recruited? Was it the control group?

The results of the study are poorly  written and incomplete

The authors are not following the general format of the discussion.. It is not discussing the findings of the study. Most of the sentences are irrelevant to the findings of the study.

Author Response

We thank the reviewers for their valuable recommendations and the attached notes. We have revised the manuscript thoroughly and hope that it is now suitable for publication

  1. The title is too general. What is meant by well-being?

Answer:

We thank the reviewer for the comment and have revised the title, we hope that it is now pointed out better.

  1. The conclusion is written as background in the abstract

Answer:

We thank the reviewer for the comment and share the opinion, therefore we have revised the background of the abstract

  1. Line 14 – How is this background related to the study?

Answer:

We thank you for the comment and agree that more background is needed.

The course was continued online during the covid pandemic in order to maintain learning progress, after the pandemic the face-to-face courses were continued.

  1. The methodology is not clear in the abstract.

Answer:

Thank you for pointing this out, we agree with the reviewer’s opinion. We have reviewed this part of the manuscript and have provided a more detailed description.

  1. Line 33- 34- this statement is about cancer survivors. How is it relevant here?

Answer:

We thank the reviewer for the comment, we have revised this part of the paper and added 2 studies that deal more explicitly with cancer patients

  1. Line 36-38 -meaning not clear

Answer:

We thank the reviewer for pointing this out and agree that this additional information is confusing and not necessarily relevant, which is why we have edited out the statement

  1. There is no supporting literature in the introduction. Most of the sentences written in the introduction is not relevant to the study. In addition, the rationale, the research gap and the aim of the study is not stated.

Answer:

We thank the reviewer for the comments and have carefully revised the introduction, emphasizing the focus of our study.

  1. The methodology is poorly written in is too brief. What is the design of the study? How were the participants recruited? How is the sample size calculated? How many participants were recruited? How the data collection was performed. What is the reason for recruiting healthy participants? How were they recruited? Was it the control group?

Answer:

We thank the reviewer for this comment. We have described that our dance course was open access, participants learn about the courses via the newspaper and  support groups. A sample was not calculated in advance.  It was a small group, so the small sample size is a limitation of our study.

A total of 51 participants took part, we show a more detailed overview in Table 1, where a distinction is also made between sex, age and patients/healthy partner. The healthy partners took part together with a partner with cancer; we did not count them as a control group, but it was noticeable that the healthy participants also experienced a short-term improvement in well-being. We also consider the small number of healthy partners to be a limitation of our study.

  1. The results of the study are poorly  written and incomplete

Answer:

We  thank the reviewer for the comments and have once again thoroughly analysed our results section.

In the results we want to deal with the physical, psychological and mental well-being of the participants before and after the training, furthermore we have dealt with the subgroup analysis in the comparison of men and women and in the comparison of patients and healthy partners and also evaluated the different well-beings in each case.

We think that we have addressed our findings, described them objectively and are not entirely clear what exactly the criticism refers to. 

The comments of the other reviewers also do not give us any indication of a possible need for correction, as this part was not seen as problematic by the other reviewers and they did not criticize the results section. 

In case there are any additional suggestions, we will be happy if we can please find out more precisely where there is still a need for correction.

  1. The authors are not following the general format of the discussion.. It is not discussing the findings of the study. Most of the sentences are irrelevant to the findings of the study.

Answer:

We  thank the reviewer for the comments and have revised the discussion. We have added further sources that support our statements. Whilst there are many studies that look t dance, there are few that look at cancer patients and few that look at ballroom dance. We have tried to point out exactly what the differences are and how they relate to our study.

Round 2

Reviewer 3 Report

Comments and Suggestions for Authors

1.      The purpose of the study is not available in the abstract section under the subheading purpose. The background is poorly written in the abstract section.

2.      The methodology in the abstract section has not been revised as per the previous comments.

3.      The introduction is too confusing, and there is no coherence between sentences and paragraphs. There is not enough supporting literature in the introduction.the rationale of the study and research gap is not available in the introduction

4.      The purpose of the study is not written properly.

5.      The methodology is not revised during revision. Study design is not available in the method. (please check my previous comments on methodology)

6.      Sample size not calculated.

7.      Which questionnaire was used for the outcome measurement? What is the reliability and validity of these questionnaires? Why the authors didn’t try for any other objective method of measurement

8.      No corrections have been made to the result section.

9.      The discussion is poorly written and often repeats points already presented in the result section. There is not enough discussion on the findings of the study.

10. In addition, I noticed a number of grammatical and typographical errors in the manuscript 
